# Engineered Immature Testicular Tissue by Electrospun Mats for Prepubertal Fertility Preservation in a Bioluminescence Imaging Transgenic Mouse Model

**DOI:** 10.3390/ijms232012145

**Published:** 2022-10-12

**Authors:** Chi-Huang Chen, Tsai-Chin Shih, Yung-Liang Liu, Yi-Jen Peng, Ya-Li Huang, Brian Shiian Chen, How Tseng

**Affiliations:** 1Department of Obstetrics and Gynecology, School of Medicine, College of Medicine, Taipei Medical University, Taipei 110, Taiwan; 2Division of Reproductive Medicine, Department of Obstetrics and Gynecology, Taipei Medical University Hospital, Taipei 110, Taiwan; 3School of Dentistry, College of Oral Medicine, Taipei Medical University, Taipei 110, Taiwan; 4Division of Infertility, Department of Obstetrics and Gynecology, Chung Shan Medical University Hospital, Taichung 40201, Taiwan; 5Department of Pathology, National Defense Medical Center, Tri-Service General Hospital, Taipei 114, Taiwan; 6Department of Public Health, School of Medicine, College of Medicine, Taipei Medical University, Taipei 11031, Taiwan; 7School of Medicine, Chung Shan Medical University, Taichung 40201, Taiwan; 8Department of Biochemistry and Molecular Cell Biology, School of Medicine, College of Medicine, Taipei Medical University, Taipei 110, Taiwan; 9Graduate Institute of Medical Sciences, College of Medicine, Taipei Medical University, Taipei 110, Taiwan; 10International Ph.D. Program for Cell Therapy and Regeneration Medicine, College of Medicine, Taipei Medical University, Taipei 110, Taiwan

**Keywords:** fertility preservation, immature testis tissue (ITT), poly-L-lactic acid (PLLA), scaffold, electrospinning, bioluminescent imaging (BLI), tissue engineering, spermatogenesis

## Abstract

Prepubertal boys with cancer may suffer from reduced fertility and maturity following gonadotoxic chemoradiotherapy. Thus, a viable method of immature testicular tissue (ITT) preservation is required in this cohort. In this study, we used poly-L-lactic acid electrospun scaffolds with two levels of fineness to support the development of ITT transplanted from transgenic donors to wild-type recipient mice. The purpose of this study was to evaluate the potential of ITT transplantation and spermatogenesis after using the two scaffolds, employing bioluminescence imaging for evaluation. The results suggest that ITT from 4-week-old mice possessed the most potential in spermatogenesis on the 70th day, together with the fine electrospun scaffolds. Moreover, bioluminescent imaging intensity was observed in recipient mice for up to 107 days, approximately six times more than the coarse electrospun scaffold and the control group. This occurs since the fine scaffold is more akin to the microenvironment of native testicular tissue as it reduces stiffness resulting from micronization and body fluid infiltration. The thermal analysis also exhibited recrystallization during the biodegradation process, which can lead to a more stable microenvironment. Overall, these findings present the prospect of fertility preservation in prepubertal males and could serve as a framework for future applications.

## 1. Introduction

Recent advances in cancer treatment have allowed the 5-year survival rate of preadolescent cancer patients to reach more than 80% in Taiwan [1]. However, cancer itself or its treatment often causes a decrease in fertility. According to past studies, young cancer patients hope to learn about the impact of cancer and cancer treatment on future fertility and the alternative fertility preservation methods before treatment. The medical team is responsible for letting patients and their guardians know the issues related to fertility preservation when diagnosed with cancer. Although sperm can be preserved by freezing following ejaculation in mature males, prepubertal boys can only consider tissue preservation to ensure fertility.

Currently, the available fertility preservation methods or those in development are mainly for prepubertal male cancer patients and those with nonmalignant hematological disorders who have been treated with radiotherapy and chemotherapy. These methods carry the risk of gonadal toxicity [2,3,4]. Most people think that the most uncontroversial way to preserve the future fertility of prepubertal boys is to safeguard the ITT and replant it after treatment. Some studies suggest that ITT in a fresh, cryopreserved, or vitrified state could preserve spermatogenesis in animals. As ITT contains self-renewing spermatogonial stem cells (SSCs), methods to preserve the tissue so that it retains its original reproductive ability after replantation have become a critical issue.

Excised ITT immediately loses the oxygen contained in the blood, various nutrients, growth factors, and biological signals supplied by the surrounding environment, which can be said to be all the biological signals required by the tissue. Therefore, when replanting the ITT, it is not easy to obtain the relevant supply of nutrients and biological signals within the required time, so most tissues cannot restore tissue function or even survive after replanting. The core strategy for immature tissue regeneration or tissue graft niche survival is to reverse the suboptimal environment caused by wounds, fibrosis, or surgical and medical interventions. The different components of the stem cell niche are complex and dynamic, requiring direct and indirect interactions among cells such as secreted factors, inflammation and scars, extracellular matrix (ECM), physical parameters, and environmental signals [5].

As we all know, the body consists of multiple organs and tissues differentiated from different germ layers. Although the functions of organs and tissues are different, hard tissues, tough tissues, soft tissues, and other tissues of varying stiffness are derived. However, the mechanical properties of cells that constitute the tiniest unit of various tissues are not significantly different. The key to creating stiffness in the organs and tissues is the stiffness of the ECM and the content of inorganic substances in the tissues. The structural order of the ECM is also crucial. Therefore, analyzing the composition and structural order of the ECM has long been one of the essential components of biomimetic native tissue [6].

As above-mentioned, the ECM is critical for nearly all tissue niches. With current advances in tissue engineering, ECM can be used as an alternative to traditional transplantation methods by creating a variety of biomaterials for use as artificial scaffolds. Enzymatically digested natural bioresorbable materials are commonly used. However, enzymatically digested natural bioresorbable materials are often unpredictable due to differences in the rate of degradation resulting from differences in the distribution of enzymes produced at individual tissue sites. Therefore, natural biomaterials such as collagen and carbohydrates that constitute the ECM in the past have readily been considered for use as bioscaffolds or cell culture matrices. Of course, good results were obtained initially. However, the high hydrophilicity of protein-based biomaterials makes it challenging to provide sufficient strength to support the scaffolds constituting the microenvironment. In contrast, polysaccharide-based materials are difficult to dissolve and fabricate due to the structure’s high density of the hydrogen bonding network.

On the other hand, the homogeneity of synthetic polymers makes their physicochemical properties easier to control, there are no antigenic problems, and they have been widely used in recent years. Among the many bioresorbable synthetic polymers, poly-L-lactic acid (PLLA) is an aliphatic polyester, one of the longest-used and most promising polymers and scaffold materials in tissue engineering research. PLLA is made to mimic the natural ECM; it is biodegradable, biocompatible, minimally inflammatory, and mechanical and it also has thermoplastic properties that make it a suitable bioresorbable implant material used in many medical applications [7,8].

To date, few studies have used PLLA-based scaffolds to facilitate ITT transplantation. Although it is well-known that the ECM both anchors stem cells and directs their fate, incomplete knowledge of the prepubertal SSCs’ niche and the conditions required for differentiation have hindered the development of suitable transplantation protocols. Furthermore, convenient and time-saving methods are lacking to elucidate human-relevant processes rapidly by tracking in vivo tissue and cellular events related to spermatogenesis or assessing the usefulness or effectiveness of alternative assays [9]. The current multidisciplinary study aims to determine the efficacy of applying fibrous, microscopic scaffolds to increase graft survival by using ITT and transgenic mice as surrogate models for isografts to mimic autologous bioengineered translational medicine transplantation. This attempt demonstrates the feasibility of the in vivo and longitudinal ITT fate mapping of spermatogenesis from sexually immature mice to adults through gonadal tissue engineering by exploiting bioluminescence imaging (BLI) (Figure 1).

BLI is a recent technology that can track animal characteristics while the animals are still alive. How can the overall research cost be significantly reduced, when the high throughput, signal acquisition time, and imaging capability are in real-time? This imaging mode, which has become common in preclinical research, is also widely used in many biomedical fields, especially genetic engineering, stem cells, and gene therapy. Our previous study also used BLI technology in transgenic mice to assess the survival status of ovarian tissue under different degrees of immunosuppressive regimens. We simulated the impact on the reproductive system of various treatments used in human infertility treatment to understand the success rate of immunosuppressive regimens [10], cryopreservation methods [11], and transplantation methods commonly used in infertility treatment research. The results showed that mice that underwent isogeneic ovarian transplantation did not show apparent rejection.

We showed that BLI technology with transgenic mice is a good monitoring platform for tissue engineering and regenerative medicine. After that, we used the technology platform again to apply it to the ITT of the male reproductive system and initially introduced the concept of tissue engineering. A fibrous structure similar to the ECM was prepared with highly biocompatible materials to establish a more suitable microenvironment for ITT. We used the average 2.64 μm polylactide electrospun mats as the carrier matrix compared with the ITT without any matrix. The study results showed that compared with the ITT without the polylactide electrospun mats, the number of days for spermatogenesis was extended from 45 to 85 days [12]. This means that a suitable scaffold can construct a microenvironment that efficiently affects ITT function. The reason for this is mainly that the structure of the electrospun mats is very similar to the fibrous structure of the ECM. Therefore, we want to extend the concept of the previous paper and prepare a structure that is closer to the testicular ECM by using a finer PLLA electrospun mat scaffold and comparing it with coarser electrospun mats. Our aim was to find a microenvironment more suitable for ITT replantation.

## 2. Results

### 2.1. Long-Term In Vivo Tracking of ITT Spermatogenesis in Age-Matched Donors and Recipients with/without Scaffold Bioengineering and BLI

We adopted the 4-week-old mice based on prepubertal age at a slow rate of spermatogenesis [13]. The surrogate model of ITT from 4-week-old age-matched mice retained better graft survival potential.

We initially compared the results of fresh ITT without a scaffold, with a coarse scaffold, and with a fine scaffold initially within the first 14 days after transplantation. The ITT image was low among the three groups, indicating reperfusion ischemia to build robust angiogenesis until the 7th to 14th day after transplantation (Figure 2A). ITT transplantation with fine scaffolds increased the BLI intensity, especially on days 14, 49, and 70 post-transplantation (* *p* < 0.05). Compared with the results of the coarse and fine PLLA scaffolds, the BLI intensity of the testicular tissue graft survival of the thin scaffolds after transplantation was greater, especially on days 1, 14, and 49 after transplantation (* *p* < 0.05). Grafted ITT reached a peak and plateau of BLI intensity from day 49 to day 70 after transplantation. From the 70th day to the 107th day, the BLI intensity gradually decreased, but the fine scaffold group showed a higher intensity than the thick scaffold group and also a much higher intensity than the fresh control group, which indicated that the fine scaffold-supported a better survival of ITT to adult testicular tissue with good reperfusion and the better restoration of angiogenesis to support the best ITT survival as early as the 14th day after transplantation. Interestingly, on day 35 post-transplantation, when mice reached early adulthood, there was no statistically significant difference, but the inclined trend of graft survival in the fine scaffold group was maintained. It is worth noting that ITT transplantation with fine electrospun scaffolds showed the highest potential at 70 days, and the BLI intensity was observed in recipient mice for up to 107 days, approximately six times more than the coarse electrospun scaffold and the control group, as shown in the bar chart (Figure 2A). The typical pattern of BLI tracked ITT grafts demonstrates the change in quantity and quality over time (Figure 2B).

### 2.2. The BLI Imaging and HE Staining of Mature Testicular Graft w/wo Scaffold

It is clear that the donor’s ITT was all from transgenic mice with strong positive luciferase between the coarse scaffold group vs. the fine coarse group and the coarse scaffold group vs. the control group. There was a significant higher photon intensity when comparing between the fine group and coarse and control group on days 14, 49, and 70 after transplantation (Figure 2A). After removal of the long-term engraftment of the mature testicular tissues from all groups, hematoxylin and eosin (HE) staining showed a typical pattern of different types of germ cells of seminiferous tubules (Figure 3), significantly more spermatid, and a larger area and circumference of seminiferous lumen in the fine scaffold group than in the control group (*p* < 0.05) (Figure 4 and Figure 5). In a comparison between the coarse scaffold group vs. the fine scaffold group and coarse scaffold group vs. the control group, there was no significant difference in the area and circumference of the seminiferous tubules in the other cell types of germ cells. Accordingly, the fine scaffold provided a smaller pore size to cope with ITT development, which provided better ECM and a niche for greater, more efficient spermatid production and a larger size of seminiferous tubules than that of the control. The coarse scaffolds may still be in suboptimal condition to improve the spermatogenesis and size of the seminiferous tubules compared to that of the control.

### 2.3. IHC and HE Staining for the Different Cells of Seminiferous Tubules

Among the IHC stains, the luciferase stain represents the donor tissue, the 3β-HSD stain represents Leydig cells [14], the OCT4 stain represents SSCs [15], SOX 9 staining represents Sertoli cells [16], and SYCP3 represents meiosis cells [17]. All of the study groups included all positive stains. In a comparison between the fine scaffold, coarse scaffold, and no scaffold, staining showed all of the intact ultrastructure with variable germ cell types related to spermatogenesis (Figure 6 and Figure 7).

### 2.4. Analysis of the Thickness of Two Scaffolds and Decellularized Testicular Tissue

The scanning electron microscopy (SEM) photos of the two electrospun scaffolds of different fineness are shown in Figure 8A,B. The micrographs show that the structure of the fine electrospun scaffolds was more like native tissue than the coarse electrospun scaffolds. Through newly prepared electrospun scaffolds that more closely resemble native tissue, we should be able to show better effects in maintaining the function of spermatogonia.

### 2.5. The Effect of Miniaturization on Stiffness

It is well-known that mammalian cells live in a porous environment. Different cell types have different pore sizes due to their different cell sizes, which has been mentioned in many papers [18,19,20]. On the issue of how to form the types of pore, as mentioned above, a fibrous electrospun scaffold similar to the native tissue was used as the basement membrane in this study. In light of the above experimental results, the fine electrospun scaffold, regardless of the luminescence performance and various activities, was better than the coarse electrospun scaffold and the control group. The reason for this may be that cells proliferate in an environment more suitable for them, and the fibrous environment that affects cell behavior is not only constructed by the biomaterial itself, but also, more importantly, the stiffness of the injection-molded forms. The stiffness of the scaffold has the same effect on cell behavior as the ECM has on cell behavior. In the past, some studies have pointed out that the cells themselves lack strength; as shown in Figure 9, various tissues had different stiffnesses, so the stiffness of the extracellular matrix in different tissues did have a huge impact on cell behavior [6,21,22]. In the past, studies using biomaterials as scaffolds for cell growth have rarely paid attention to the problem of matching the stiffness of the scaffold with the target tissue. The aim of this study was to reconstruct part of the testicular tissue, so it is important to understand the stiffness of the tissue as the target. It is known that the stiffness of the human testicular tissue is approximately 4.77 ± 1.16 kPa [23], and this study used PLLA, a highly biocompatible material that is suitable as a material for biological scaffolds. The stiffness of bulk PLLA and commonly used plastics is 2.59 GPa [24] and 2.7–4.14 GPa [25], respectively. In addition, the entangled status of the polymer chains fabricated by processing will change their stiffness. For example, electrospinning is a process for nanonization that also causes a dramatic decrease in stiffness, leading to it being easy to cause changes in cell behavior. The stiffness of the electrospun scaffold was reduced from the GPa level to 88 ± 22 MPa at the MPa level, and the stiffness was further reduced to 800 ± 100 kPa after being treated with alcohol immersion [26]. In other words, the stiffness reduction depends on the fineness increase, and body fluid infiltration will reduce its stiffness. Thus, it can be seen that the fine electrospun scaffold should be closer to the stiffness of the native testicular tissue, which is the preliminary reason why we speculated that the fine electrospun scaffold had better performance than the others.

### 2.6. The Scaffold Material, Fixed Mass, and Stable Production Process

Furthermore, we used the same material, fixed mass, and stable production process. Injection molding was used to conduct *in vitro* experiments at 37 °C in phosphate buffer solution (PBS) to simulate *in vivo* degradation behavior experiments. The investigation was carried out for a total of 52 weeks, as described in Material and Method, and the appearance of the molded article after 26 weeks is shown in Figure 10. To Understand the Dimensional Stability of (5D/95L) PLLA in the Simulated Body Fluid Environment, We also Added the (15D/85L) PLLA Molding to the Experiment for Comparison The Comparison of the Thermal Analysis Results of Glass Transition Temperature (Tg), Crystallization Temperature (Tc), and Melting Temperature (Tm) The Results of PLLA Hydrolysis, Thermal Analysis, and Gel Permeation Chromatography Measurements. The Mass (Red Marker and Line) Will Increase Slightly with the Degradation of PLLA.

## 3. Discussion

With the successful advent of advanced techniques to develop biomaterials applied to biomedicine including variable organs or tissues, we aimed to conduct a series of studies on engineered ITT transplantation. Following our previous study [12], the PLLA scaffolds can be used to analyze the biocompatibility, biodegradation, and mineralization during ITT transplantation into adulthood. Many studies have shown that microscopically, in addition to dynamic factors such as endocrine influences and blood flow, which affect the fate of cells and the reconstruction of tissues, the slower changing environment of physical characteristics such as biocompatible metals, ceramics, or nondegradable polymers show fewer changes in the short-term, but the impact is long-term [5]. To date, using porous scaffolds to simulate the ECM in the tissue provides cells with a 3D scaffold for tissue reconstruction. However, fabricating a scaffold that can satisfy all parts and tissues is challenging under the mutual influence of strength, porosity, permeability, and other factors.

In response to the countermeasures developed by the existing male reproductive preservation strategy, we can use a simple method to decellularize the tissue to understand the fine structure of the extracellular matrices in the tissue as the basis for artificial scaffold simulation. It can also be used directly as a substitute for allogeneic or xenogeneic scaffolds. In the past, several works have been carried out related to this phenomenon [27,28]. Therefore, we used sodium dodecyl sulfate (SDS) to decellularize testicular albuginea and observed the decellularized scaffold (i.e., the extracellular matrix). The SEM in Figure 8C shows the scanning electron micrograph of decellularized testicular albuginea with an apparent fibrous network structure; the fiber fineness was approximately 150 nm. Furthermore, electrospinning is undoubtedly the best choice for simulating the fibrous porous state in terms of the existing fabrication technology; the single processing procedure of electrospinning makes it relatively simple to perform in the laboratory. We also conducted preliminary research on the native testicular albuginea structure by electrospinning [12]. The fibrous network structure fabricated by electrospinning was similar to the native decellularized testicular albuginea structure. Its high permeability allows for blood flow, nutrients, and other biosubstrate supplies to be delivered in a timely way. Therefore, we fabricated two kinds of electrospun scaffolds with different fineness in this study and the transplanted scaffolds with chopped testicular tissue and measured the response using IVIS. The schematic experimental process is shown in Figure 1.

In addition to the physical properties of the initially used scaffolds that affect the stiffness, another factor that must be considered to affect the changes in the cell growth environment is the degradation behavior of biodegradable polymers. The space vacated by the degradable macromolecules after being biodegraded and metabolized will allow regenerated cells and interstitial tissue to refill the space generated by the degradation and make up for the strength. Therefore, biodegradable polymers as scaffolds for regeneration are a dynamic equilibrium in the process of tissue reconstruction. Among them, the degradation rate of the degradable polymer is also affected. On the other hand, when the biodegradable biomaterial is biologically inactive at the high molecular weight stage, after hydrolysis, the biodegradable polymer will gradually become oligomers or monomers and become bioreactive and enter the metabolism to affect the body. The processing method and crystallization field of degradable biopolymer scaffolds will affect the degradation rate and thus the number of single molecules impacting the body. These factors will also affect the stiffness of the scaffold and, therefore, change the cell behavior. Consequently, we followed another series of in vitro degradation experiments to understand the changes in the degradation process of our PLLA fine electrospun scaffolds in vivo. The PLLA used as this study’s electrospun scaffold was composed of 5% D-form and 95% L-form PLLA. However, due to the low density and loose shape of the finished electrospun scaffold, it is not easy to explore the degradation behavior of PLLA.

In Figure 11 and Figure 12, the Tg, Tc, and Tm started to appear at around 61, 121, and 150 °C, respectively, and the temperature change trends are shown in green (a), red (b), blue (c), and (d) arrow representation. The results show that the change in Tg was minimal, and the Tg changed little after 52 weeks of hydrolysis. Looking at the change in Tm, we also found that from the initial 150 °C, the main chain in PLLA was hydrolyzed to generate random cleavage. After that, there were two groups of main chain groups with different molecular weights, and there were two major peaks of Tm, namely, Tm1 and Tm2. The change in Tm was not significant. In light of the thermal analysis results, it is worth noting that the crystallization temperature, Tc, was low in D-form content. PLLA is a kind of polymer that is easy to crystallize. Therefore, we found that the initial crystallization temperature was moderately endothermic, around 121 °C, from the rapidly decreasing injection molding. At the peak (endothermic peak), with the progress of hydrolysis, the crystallization temperature shifted to the low-temperature side, and the peak shape became sharper. In the 52-week sample, Tc shifted to 95 °C, which means that the PLLA molecular chain after hydrolysis became shorter. This made it easier to form crystalline regions and complemented the rigidity reduced by shortening the molecular chain.

In contrast, the molecular weight (green square marker and line) continued to decrease with the increase in degradation, from the initial Mw = 148 kDa to Mw = 29.5 kDa after only 52 weeks. In general, the whole degradation time of PLLA dense moldings with a molecular weight of approximately 150,000 Da takes around 5–7 years [29]. The presence of PLLA fibers was also found in our 107-day tissue sections. Although theoretically the change in strength (green circle marker and line) should decrease with a decreasing molecular weight, significant strength delays of up to 10 weeks could be found in Figure 13 [30,31].

Furthermore, the degree of crystallinity (blue marker and line) gradually increased from the initial 7.6% with the extension of the hydrolysis time and grew to approximately 25% when approaching 52 weeks. A similar result could also be observed in the change in Tc in Figure 11. The main chain of PLLA was cleaved by hydrolysis; the cleaved main chain with a smaller molecular weight could more quickly enter the crystallization regions to make up the strength and stiffness that are due to the reduced molecular weight that accompanies strength and stiffness losses. This phenomenon can lead the microenvironment of the molding form to be more stable. Therefore, we can infer that the fiber miniaturization of fine scaffolds was higher than that of the coarse scaffolds, making their stiffness more suitable for reconstructing testicular tissue.

There were several drawbacks in this study. BLI signals could represent tissue viability and is probably related to spermatogenesis, but not the stages of cell differentiation. The HE staining provides limited information and lacks accuracy in differentiating germ cell types, and our limited cell marker for partial germ cell types of spermatogenesis is not able to present the full information on the real number of different types of germ cells. Additional cell markers to analyze the different types of germ cells such as DDX4, ZBTB 16, PLZF, POU5F1, STRA 8, KIT, SYCP3, DAZL, SALL4, TRA 98, DPPA3, and STELLA deserve further study. The results were also derived from the 4-week-old age-matched donors, and recipients for germ cell proliferation remained at the prepubertal stage at a slow rate in spite of spermatogenesis beginning as early as 1.5 postnatal days [13]. We applied the in vivo BLI tracking system to 4-week-old mice, which is equal to the pre-pubertal age, with germ cell proliferation at a slow rate, the functional maturation of Sertoli cells, and the production of functional Leydig cells [13]. Finally, we did not prove the fertility outcome to produce offspring from the transplanted ITT. More studies are required to produce viable offspring after the adoption of this research model as a translational study for human application.

Combined biomaterials such as PLLA scaffold application during ITT transplantation by an in vivo BLI seems to have promising potential for multidisciplinary research on the fertility preservation of prepubertal boys with cancer or hematologic disorders.

## 4. Materials and Methods

### 4.1. Preparation and Characterization of Scaffolds

#### 4.1.1. Materials

Two kinds of PLA, (5D/95L)PLA and (15D/85L)PLA, with different levorotatory/dextrorotatory ratios of lactide, were used. Poly 5D/95L-Lactide ((5D/95L)PLA, Mw = 140 kDa, Tg = 60–62 °C), was purchased from Biotech One Inc. (New Taipei City, Taiwan), and Poly 15D/85L-Lactide ((15D/85L)PLA, Mw = 164 kDa Tg = 57–60 °C) was purchased from MacroPore Biosurgery, Inc. (San Diego, CA, USA). *N*,*N*-Dimethylformamide (DMF) was distilled over CaH_2_ at a low pressure under a nitrogen atmosphere and stored over Linde-type 4A molecular sieves. All chemicals purchased were used as received without further purification

#### 4.1.2. Electrospinning of PLLA

(5D/95L)PLA was dissolved in dichloromethane (CH_2_Cl_2_) or a solvent mixture of CH_2_Cl_2_ and DMF at 8:2 (*v*/*v*). The electrospinning conditions were as follows. The delivery capillary had an inner diameter of 1 mm. A DC motor with a 12.5 cm diameter, composed of a stainless-steel disc, served as a collector and was located approximately 15 cm away from the capillary tip. The applied voltages for the positive and negative charges were 7–22 kV and 2–8 kV, respectively.

#### 4.1.3. Decellularization of Testicular Tissue

Fresh tunica albuginea from the FVB/NJNarl wild-type mice was harvested. The cleared testicular tissue was placed in a lyophilizer (Kingtech, New Taipei City, Taiwan). The settings of the lyophilizer were fixed at –80 °C and 100 mTorr and left overnight. After lyophilization, the lyophilized tunica albuginea was placed in 200 mL of 1% SDS at room temperature. The detergent solution was changed twice every two weeks and gently shaken during decellularization. After decellularization, the testicular tissue was lyophilized again under the same conditions above-mentioned.

#### 4.1.4. Scanning Electron Micrograph

The morphology of the obtained fibrous scaffolds and the decellularized testicular tissue were observed under SEM (S-2400, Hitachi, Japan) at an accelerating voltage of 15 kV.

### 4.2. Animal Studies

#### 4.2.1. Transgenic Mice

Mouse testicular tissue donors were of a transgenic mouse line (FVB/N-Tg (*PolII-Luc*) Ltc transgenic mice with H2q haploid genotype) created by the Level Transgenic Center of Level Biotechnology Inc. (Taiwan). Figure 14 presents a typical representation of a BLI application, a transgenic construct of the promoter and the modified firefly luciferase gene, and a PCR genotyping of the transgene. Figure 14A shows the PCR genotyping of the transgene. Figure 14B shows a pre- (transgenic mice 1 and 2) and post- (wild-type; mouse 3) in vivo BLI photograph of the transgenic mice and testicular tissue after injection or infusion with luciferin. The cell plate shown in Figure 14B contained testicular tissue removed from mice numbered 1, 2, and 3. The testicular tissue from mouse 1 was released after the intraperitoneal (i.p.) injection of luciferin, whereas tissue from mouse 2 was first removed and then infused with luciferin. The construction of the targeting gene fragment RNA Polymerase II promoter (*PolII*) with a modified firefly luciferase cDNA (pGL-2; Promega, WI, USA), and the generation and maintenance of the murine germline have been described in our previous articles [9,11,32]. Recipient mice were an inbred strain of FVB/NJNarl wild-type mice (National Laboratory Animal Center, Taiwan) with the H-2 haplotype (H2q).

Housing and breeding conditions in the animal house for the mice were under 22–24 °C and followed a 12/12 h light/dark regimen without the restriction of food and water supplies. The Animal Experimental Committee at the Taipei Medical University (Taipei, Taiwan) reviewed and approved the above-mentioned procedures that adhered to the Guide for the Care and Use of Laboratory Animals (National Institute of Health).

#### 4.2.2. Implantation of Testicular Tissue

This study was based on a preliminary experiment in our previous article [12]. The experiment first confirmed that testicular tissue is one of the best sites of the luminescence response in the tissue regeneration of mice. This study mainly followed the results of that study. The recipient transplanted age-matched donor testicular tissue to the scrotum of the orchiectomized animals. The difference in the implantation was that the donor’s testicular tissue used two types of PLLA scaffold: coarse and fine. Mice directly implanted without a scaffold were used as the control group (Figure 2). FVB/N-Tg (*PolII-luc*) Ltc transgenic male mice were used as testicular tissue donors, and inbred FVB/NJNarl wild-type mice were used as recipients. All operations and transplantation procedures were performed under general anesthesia induced by i.p. ketamine (50 mg/kg) and xylazine (15 mg/kg). One week before each experiment, the recipient mice underwent bilateral orchiectomy.

#### 4.2.3. Bioluminescence Imaging In Vivo

Mice testicular tissue grafts were longitudinally tracked by BLI with the IVIS-200 system, and the luminescence was quantified using IVIS software. Nude skin mice are preferable due to the autofluorescence of fur that may interfere with the intensity of the bioluminescence signal. Therefore, the skin was shaved 2 h before imaging. Luciferin (150 mg/kg) was injected i.p. into mice 10 min before imaging, and recipients were subsequently placed in a light-tight camera box under continuous isoflurane (2%) general anesthesia. Mice were imaged from the ventral side with a field of view of 20 cm for 3 min at high-resolution settings. A cooled charge-coupled device (CCD) camera was used to perform the BLI with an additional overlay image of a black-and-white picture taken with the aid of a light in the imaging chamber.

Luminescence was quantified by summing the pixel intensities inside the region of interest (ROI; 1.5 cm × 1.5 cm) and absolute light intensity calibrated using an 8-inch integrating sphere (OL series 425 Variable Low-Light-Level Calibration Standard, Optronic Laboratories, Inc., Orlando, FL, USA), as described in [33]. Light emitted by the catalytic reaction of luciferin by luciferase within the ROI was measured by the photon counts on the digitized image captured by the CCD camera on the integrating sphere. The photon counts were converted to physical units of radiance in photons/(s per cm^2^ per steradian) [34]. IVIS software was used to quantify and archive signals from the images. In addition, background signals were evaluated daily to perform background subtraction calculations. Digitized BLI images and photon intensities were initially recorded 1 day postimplantation, followed by weekly recordings. However, on some occasions, due to the lack of availability of BLI equipment, some data record dates differed by 1 to 2 days.

#### 4.2.4. Histological Observation

At the end of the study, testicular grafts from the recipient mice were removed for histological studies. The testicular tissues were fixed in 10% formalin overnight, embedded in paraffin wax, sectioned into five micrometer (5 μm) intervals, then dewaxed and stained with HE. Randomized histological sections were studied from each recipient (magnification of 2× and 20×). Native testis served as the control and grafted testicular tissue with PLLA or without the scaffold (tissue-only) to examine the donor grafts from transgenic mice, the intact morphology of seminiferous tubules, and the presence of germ cells.

A Nikon ECLIPSE Ni (Japan, Tokyo) microscope and Nikon digital sight DS-U3 camera (Japan, Tokyo) were used to measure the cell counting of different cell types of germ cells under 200×.

#### 4.2.5. Immunohistochemical (IHC) Staining

Next, 5 μm-thick, Bouin-fixed paraffin-embedded tissues were used for IHC staining as described [10]. Immunohistochemical analysis was performed using Leica BOND-MAX™ (Leica Biosystems, Wetzlar, Germany) using the peroxidase method with antibodies to luciferase (1:1000, Cambridge, UK), 3B-HSD (1:200, Abclonal, New Taipei City, Taiwan), SOX9 (1:300, Abclonal, New Taipei City, Taiwan), Oct4 (1:100, Abclonal, New Taipei City, Taiwan), and SYCP3 (1:300, Novus Biologicals, Colorado, USA). According to the company’s recommended procedure, the Leica BOND-MAX avidin-biotin-free polymer system was used in the detection. The slides were counterstained with HE. The expression of firefly luciferase, 3B-HSD, SOX9, Oct4, and SYCP3 was evaluated under a slide scanner MoticEasyScan Pro 6 with Motic DSAssistant software (Motic China; Xiamen, China).

#### 4.2.6. Statistical Analysis

A one-tailed *t* test was used for the statistical analysis of BLI intensity and *p* < 0.05 was considered statistically significant.

For the statistical analysis of the size of seminiferous tubules and different types of germ cells, the response variable of interest is not normally distributed; we performed the Kruskal–Wallis test to test for differences between treatments and the Nemenyi test for post hoc comparison [35].

### 4.3. Degradation Behavior

#### 4.3.1. Preparation of PLLA Molding

PLA molding was compression molded into a bar shape of 25 mm × 6 mm × 3 mm (bar shape) and a Ø15 mm × 1 mm (disc shape) cubic size molding.

#### 4.3.2. Hydrolytic Degradation

Each PLLA molding sample was placed into individual 20 mL sample vials for the PLA hydrolysis experiment. All hydrolyses were performed in PBS and followed the ASTM F1635-95 (2000) standard test method for the in vitro testing of PLLA. In brief, the samples were initially placed under a reduced pressure at room temperature for 24 h to remove excess water. Vials were then prepared to contain one compression-molded PLA with 10 mL PBS. The sealed sample vials were then placed into a 37 °C horizontal shaking water bath with 30 rpm for 0–52 weeks. The PLLA samples were removed from PBS at each predesignated time and gently blotted with a KimWipe^®^. They were then dried at a reduced pressure (25 torr) at room temperature for 24 h. All hydrolytic degradation experiment samples were run in triplicate.

The weight change was calculated according to the following equation:$$\mathrm{weight}\;\mathrm{loss}\;\left(\%\right)=\frac{w_{0}-w_{t}}{w_{0}}\;\times \;100$$
where *w*_0_ and *w_t_* represent the initial weight and the weight at time *t*, respectively.

#### 4.3.3. Characterization of the Thermal Behavior of Hydrolyzed PLA Moldings

The thermal behavior analysis of hydrolyzed PLLA moldings was performed using DSC (Pyris 1, Perkin-Elmer, Waltham, MA, USA). Approximately 5 mg of the sample was placed in an aluminum pan. The DSC scans were obtained during the following heating process: the samples were heated from 30 °C to 220 °C at a rate of 10 °C/min, maintained at 220 °C for 5 min, and cooled from 220 °C to 30 °C at a rate of 50 °C/min. The analysis of DSC curves for the heating process was carried out for the second heating data. For all experiments, the heating chamber was protected by nitrogen at a flow rate of 25 mL/min.

## 5. Conclusions

In this study, the fine electrospun scaffold composed of finer PLLA fibers constituted an environment for reconstructing testicular tissue for the fertility preservation of prepubertal boys, and it was found that the photon activity was approximately six times higher than that of the coarse PLLA electrospun scaffold we used in the past, indicating that a more suitable microenvironment was established. A more appropriate reason for this could be that the micronized PLLA fibers make the overall microenvironment more closely resemble the original testicular tissue. The degradable PLLA also stabilizes the microenvironment due to recrystallization. Before ITT engineering through scaffolds can be established as a new approach in spermatogenesis for the fertility preservation of boys with prepubertal cancer or hematologic disorders, the safety of material sciences must be ensured.

## Figures and Tables

**Figure 1 ijms-23-12145-f001:**
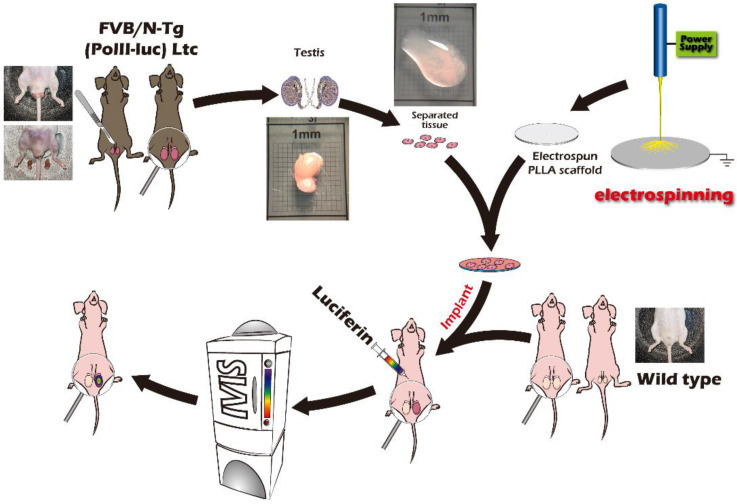
Schematic overview of the study design.

**Figure 2 ijms-23-12145-f002:**
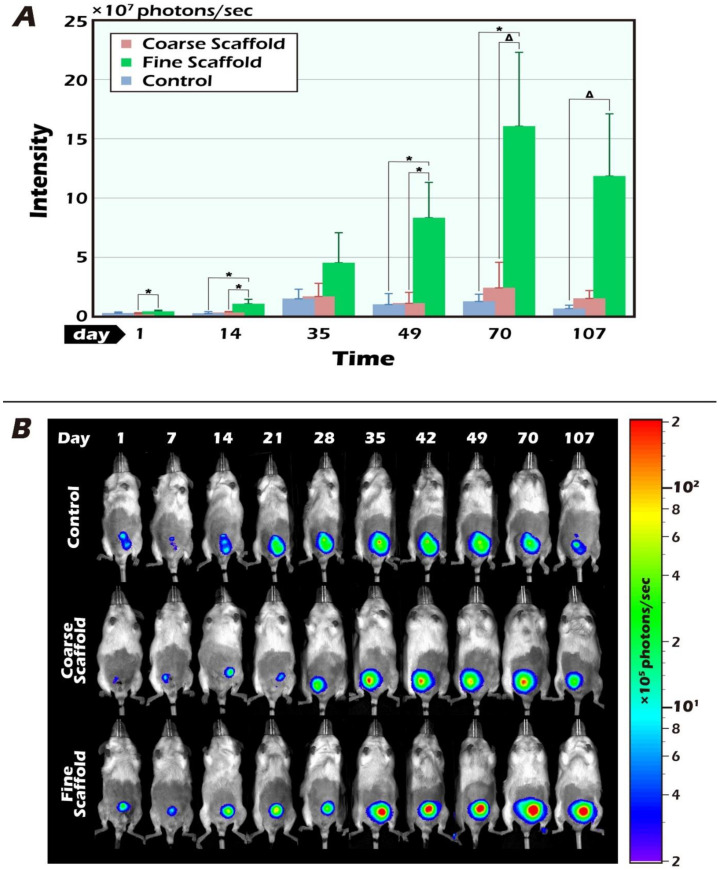
(**A**) Longitudinal bioluminescent imaging of luciferase activation. Four-week-old recipients, where the transplanted donor testicular tissue was age-matched to the scrotum of bilateral orchiectomy with coarse, fine, or without an electrospun PLLA mat. The grafted testis tissue with an electrospun PLLA mat could increase the degree of cell regeneration compared with and without the scaffold, in particular on days 14, 49, 70, and 107 after transplantation when compared between the fine scaffold group and control group (* *p* < 0.05, Δ *p* = 0.05). Based on the BLI quantity, the signal intensity of the immature testicular graft survival increased to the crucial peak between the 49th and 70th days and then decreased at the end of the 107th tracking day. The quantity of BLI showed significantly effective but peak BLI intensity with the fine electrospun PLLA mat. (**B**) Representative pattern of in vivo tracking of the BLI image of a typical in situ ITT engraftment in an individual.

**Figure 3 ijms-23-12145-f003:**
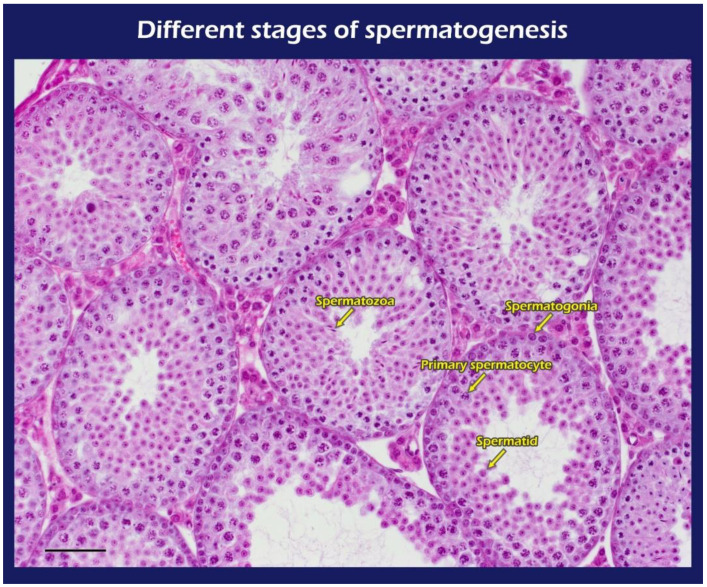
Typical different types of germ cells during spermatogenesis including spermatogonia, primary spermatocyte, spermatid, and spermatozoa (yellow arrow). Bar = 60 um.

**Figure 4 ijms-23-12145-f004:**
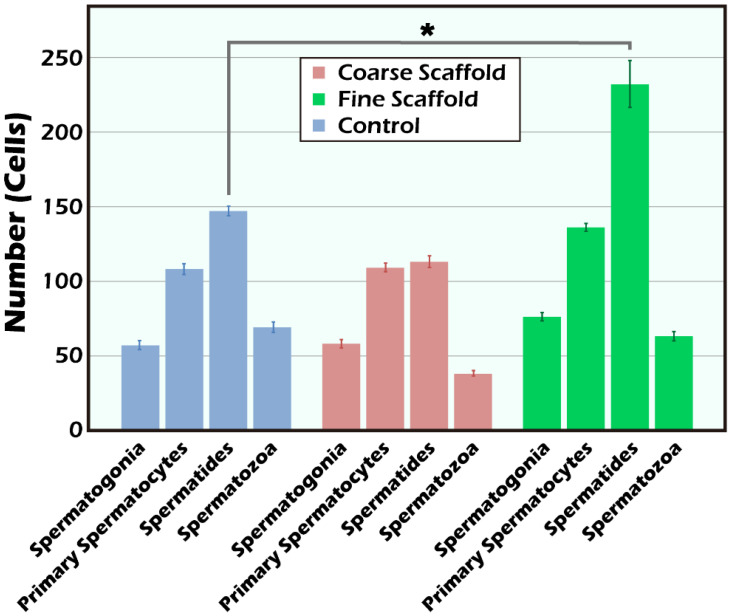
The cell counting of different types of germ cells in the seminiferous tubules were measured and quantified by their distinct morphology including all spermatogonia, spermatocytes, spermatids, and spermatozoa under HE staining 200×. Comparing the difference between different types of germ cells among each group, the fine scaffold group showed superiority in the cell count of spermatids in comparison to the control group (* *p* < 0.05). In the coarse scaffold, fine scaffold, and control groups, there were no significant differences in the percentages of spermatogonia, primary spermatocytes, or spermatozoa.

**Figure 5 ijms-23-12145-f005:**
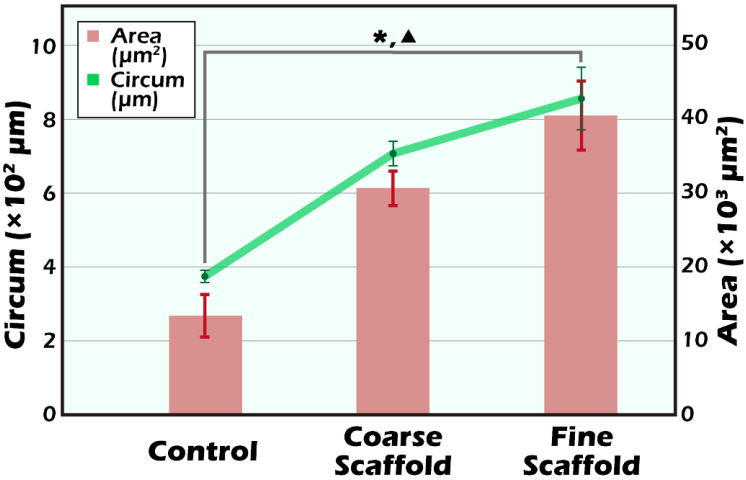
The measurement of the circumference and area of seminiferous tubules was randomly applied to three seminiferous tubules (mean +/SD) and displayed a significantly larger area in the fine scaffolds in comparison to the control group (* *p* < 0.05). With regard to the circumference of seminiferous tubules, the fine scaffolds also showed better results as opposed to the control group (▲ *p* < 0.05). There was no significant difference between the coarse scaffold and fine scaffold, or the coarse scaffold and control groups.

**Figure 6 ijms-23-12145-f006:**
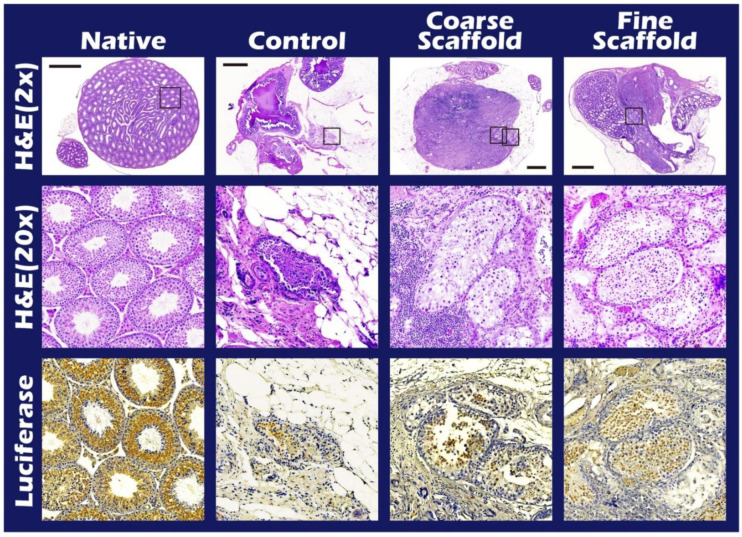
Representative images of testicular tissue with/without PLLA mats after preliminary HE and immunohistochemical (IHC) staining. The HE stain shows the existence of spermatogenesis in the order of the fine electrospun PLLA mat group (row 2), coarse electrospun PLLA mat group (row 2), and the control group (row 2) after long-term engraftment of the donor mature testicular tissue. The IHC stain correctly showed the testicular tissue of the FVB/N-Tg (*PolII-luc*) Ltc transgenic donor in four groups (row 3). The HE images in the first and second row from 2× to 20× and the IHC image in the third row of the box.

**Figure 7 ijms-23-12145-f007:**
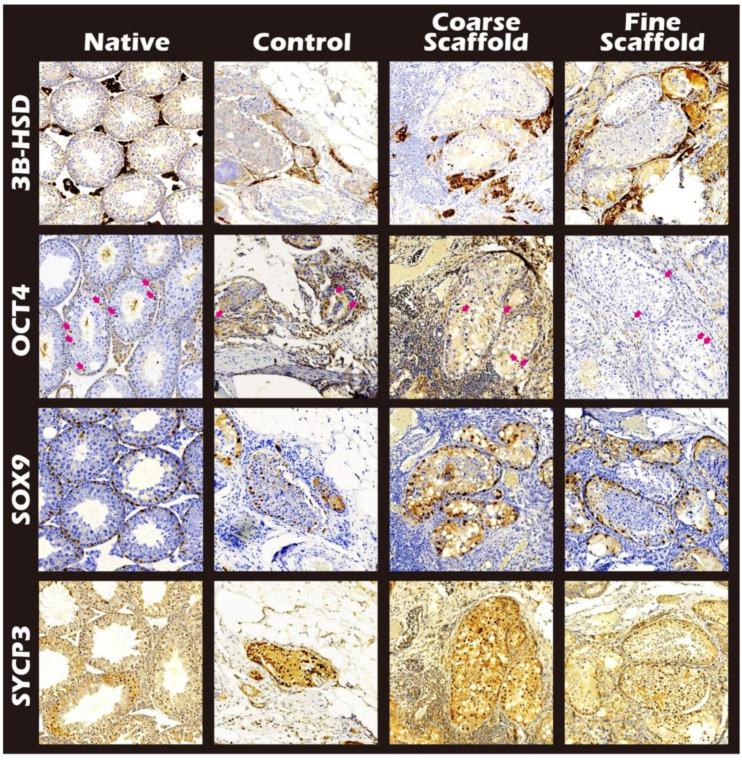
Various functional immunohistochemical staining images of the donor mature testicular tissue within different fineness of PLLA electrospun mats after long-term engraftment. In the images, the first line is the native tissue, and the second to fourth lines are the donor mature testicular tissue encapsulated for 107 days without scaffolds (line 2), the coarse PLLA electrospun mats (line 3), and the fine PLLA electrospun mats (line 4). Positive 3B-HSD stain for Leydig cells (row 1), positive OCT4 stain for spermatogonial stem cells (arrows, row 2), positive SOX9 stain for Sertoli cells (row 3), and the SYCP3 stain (row 4) was required for meiosis during spermatogenesis in all groups, but the existence of spermatogenesis is shown in order of the fine scaffold, coarse scaffold, and control group (Figure 6).

**Figure 8 ijms-23-12145-f008:**
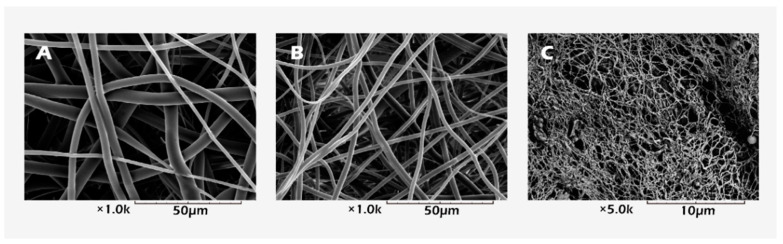
SEM images of the (**A**) coarse, (**B**) fine (1000× magnification) electrospun PLLA mats, and (**C**) decellularized testicular tissue (5000× magnification).

**Figure 9 ijms-23-12145-f009:**
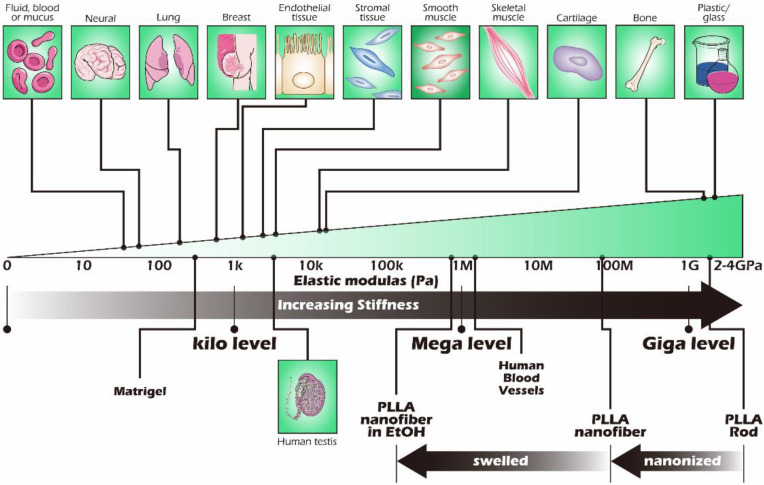
Biomechanical characteristics of mammalian tissues (variations in tissue stiffness). The biomechanical properties of tissue in terms of stiffness (elastic modulus) are usually measured in Pa. There are differences in stiffness between the organs and tissues, and this is closely related to tissue function in nature. Most of the soft tissues are lower than the kPa level, and the hard tissues are as high as the GPa level; the bioscaffold plays a key role as an extracellular matrix to create an environment similar to the original tissue to facilitate the reconstruction and regeneration of the native tissue. Reproductive system tissues are approximately 1–5 kPa. This schematic illustration was redrawn after referring to the paper by Thomas R. Cox and Janine T. Erler [21] and adding our concepts.

**Figure 10 ijms-23-12145-f010:**
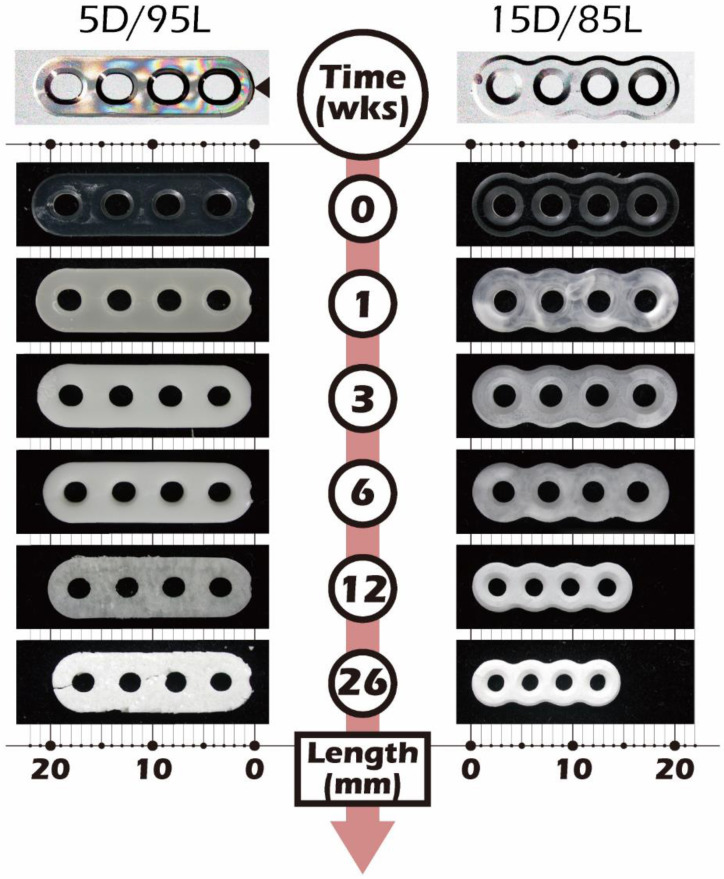
Changes in different PLLA forms with 5% and 15% dextrorotatory ratio after hydrolysis in PBS for 1–26 weeks. The left images are 5D/95L-PLA forms made by molding injection, and the right photos are 15D/85L-PLA forms made by extrusion and milling.

**Figure 11 ijms-23-12145-f011:**
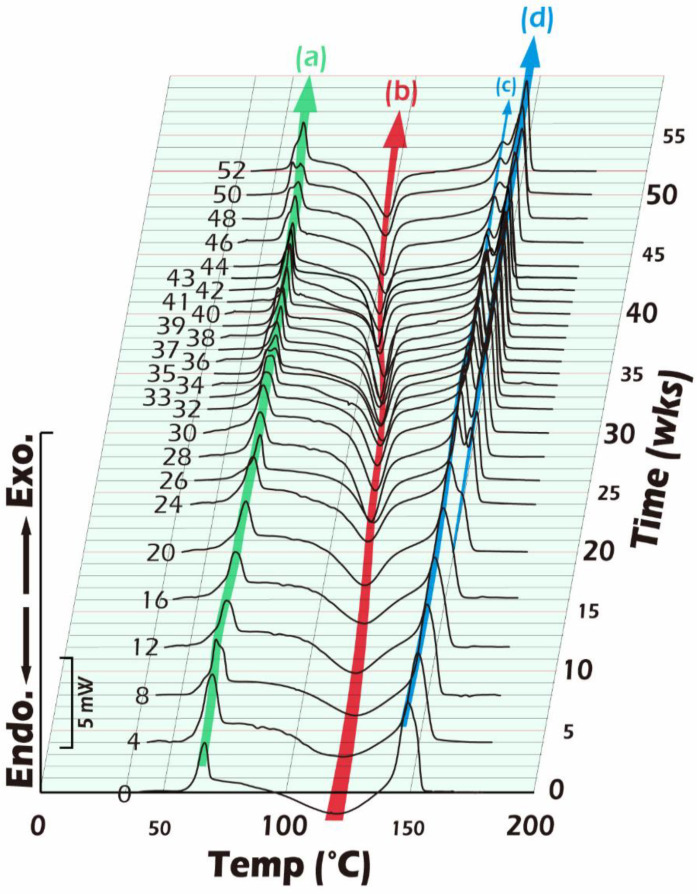
The 3D DSC thermograms of partially crystalline PLLA samples show the presence of the recrystallization phenomenon and melting shifting after hydrolysis in PBS for 1–52 weeks at 37 °C.

**Figure 12 ijms-23-12145-f012:**
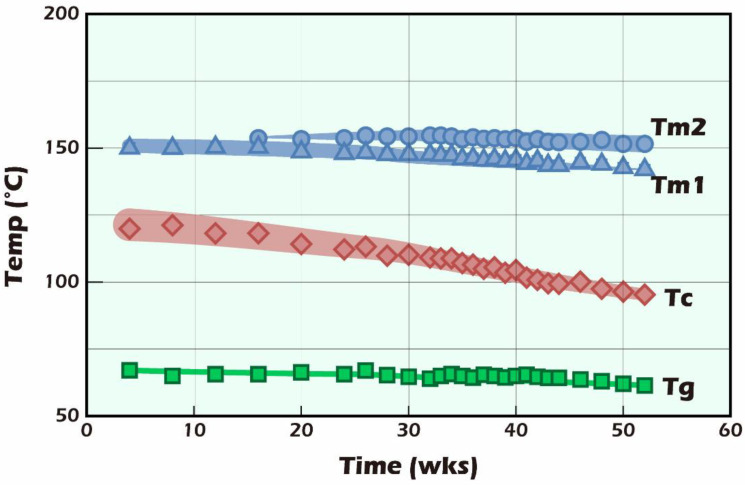
The variety of partially crystalline PLLA samples in the temperature of glassification (Tg), crystallization (Tc), and melting (Tm) after hydrolysis in PBS for 1–52 weeks.

**Figure 13 ijms-23-12145-f013:**
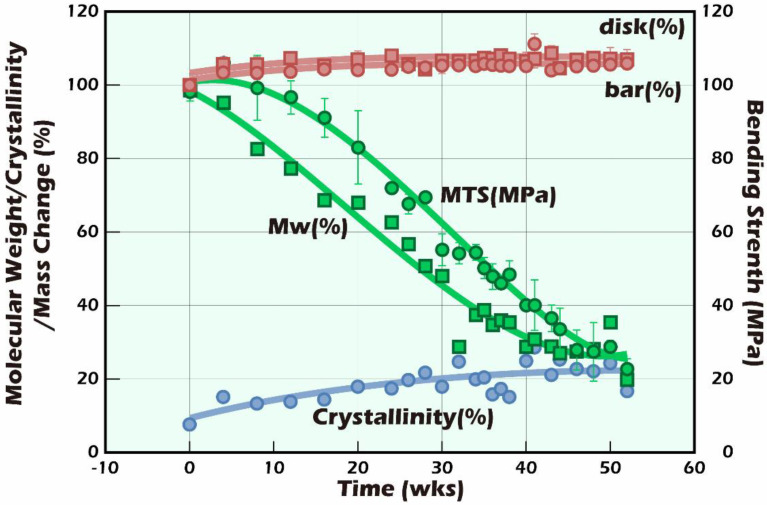
The variation in the partially crystalline PLLA samples in bending strength (MPa), crystallinity (%), and weight-average molecular weight (Mw) after hydrolysis in PBS for 1–52 weeks. MTS tested the bending strength; the gel permeation chromatography (GPC) measured Mw; and the Tc area of the DSC thermograms was used to calculate the crystallinity.

**Figure 14 ijms-23-12145-f014:**
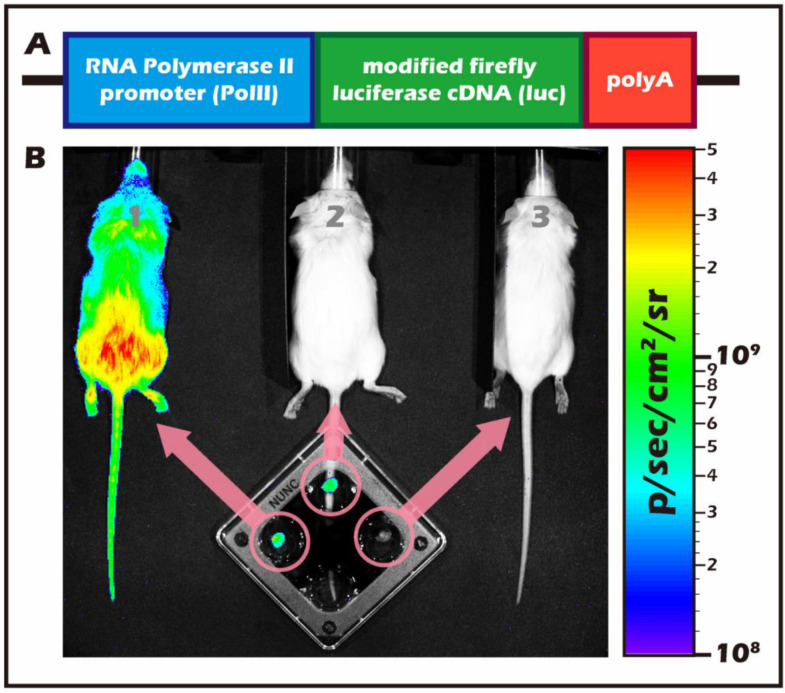
(**A**) Schematic design of the transgenic construct including the 712 bp mouse RNA Polymerase II (*PolII*) promoter and a modified firefly luciferase cDNA (Promega pGL-2). (**B**) FVB/N-Tg (*PolII-luc*) Ltc transgenic donor mice (left-1) were treated with luciferin; the light emitted by the transgene reaction was detected with the IVIS imaging system. The luciferase expression was detectable in all organs of the transgenic mouse including the isolation of intact testis independently (middle-2). No light emission was detected from a wild-type donor intact testis isolated from the recipient mice (right-3).

## Data Availability

All data generated in the study are presented in the manuscript.

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
