# Peer review of "Engineered Immature Testicular Tissue by Electrospun Mats for Prepubertal Fertility Preservation in a Bioluminescence Imaging Transgenic Mouse Model"

_ijms, 2022, doi:10.3390/ijms232012145_

Round 1

Reviewer 1 Report

In this study, the authors used poly-L-lactic acid electrospun scaffolds with two levels of fineness to support immature testicular tissue (ITT) transplanted from transgenic donors to wild-type recipient mice. They reported in a previous study that spermatogenesis could be extended from 45 to 85 days when using a polylactide electrospun scaffold. The purpose of the present study was to prepare a structure that is closer to the testicular extracellular matrix and to quantitatively evaluate ITT transplantation and spermatogenesis using bioluminescence imaging, hematoxylin-eosin and immunohistochemical (IHC) staining.

According to the authors, the best spermatogenesis potential was observed at 70 days with fine electrospun scaffolds and spermatogenesis was observed in recipient mice for up to 107 days, about 6 times more than in the “coarse electrospun scaffold” group and the control (without scaffold) group. Their thermal analyses showed that recrystallization occurs during the biodegradation process, which can lead to a more stable microenvironment. The authors conclude that their results can serve as a framework for further research on fertility preservation in prepubertal males.

Although the aim of this study is of interest, the work presented is too preliminary for publication. The results section is not clearly written and the manuscript is not well structured (please see below). Importantly, the conclusions on spermatogenesis are not supported by the results (please see below).

-  Materials and Methods are found in the Results section.

-  Results are found in the Discussion section.

-  The discussion is too long and not a real discussion.

-  Many sentences, paragraphs and legends are unclear, confusing or misleading in the manuscript. The English writing should be greatly improved.

-  The two graphs presented in figure 2 are redundant. Moreover, figures 2 and 3 should be regrouped in the same figure.

-  The authors used four-week-old transgenic donors and do not justify this choice. Indeed, in prepubertal boys, ITT (in which the only germ cells are spermatogonia) is preserved for fertility preservation. In four-week-old mouse testes however, several types of germ cells can be found: spermatogonia, spermatocytes and spermatids.

-  Lines 206-208 “Significantly thicker seminiferous tubules in the scaffold groups and a larger seminiferous lumen with significant spermatogenesis than that of ITT without a scaffold”

No measurements of the diameter of seminiferous tubules, of the height of the seminiferous epithelium, of the lumen area and no statistical analyses of these data are shown in this manuscript.

-  Lines 2018-219 “The HE stain shows significantly adequate spermatogenesis” The different types of germ cells cannot be distinguished at this magnification. Moreover, the percentage of the different types of germ cells (spermatogonia, spermatocytes, spermatids and spermatozoa) per seminiferous tubule should be determined and compared between the different experimental conditions.

-  Line 225 “DDX4 represents differentiated germ cells” DDX4 is expressed in all the types of germ cells and not restricted to differentiated germ cells. Other IHC experiments should be performed to detect specifically differentiated germ cells.

-  Lines 232-233 “The pore size of the fine scaffold showed the best ECM-like niche to engineer ITT transplantation followed by the best spermatogenesis” No quantitative data on the progression of spermatogenesis are shown in the manuscript.

-  The same IHC images showing the expression of luciferase are presented in figures 4 and 5.

-  Figure 5: DDX4 staining is more intense in the interstitial tissue, and not within seminiferous tubules as expected.

-  Figure 5: OCT4-positive cells are not visible at this magnification.

-  Figure 5: SYCP3 staining is very diffuse and does not seem to specifically stain the spermatocytes.

-  No quantification and no statistical analyses of the different immunostaining presented in figure 5 are shown.

-  Line 229: “variable degree to achieve spermatogenesis” No quantification and no achievement of spermatogenesis are shown in this manuscript.

-  Lines 230-231 “fine scaffold group > course scaffold group > control group” No quantification and no statistical analyses were performed. How can the authors draw this conclusion?

-  Lines 244-245 “significantly better spermatogenesis in order as the fine scaffold, coarse scaffold, and control group.” Same comment as above.

Author Response

Response to Reviewer 1

Comments and Suggestions for Authors

In this study, the authors used poly-L-lactic acid electrospun scaffolds with two levels of fineness to support immature testicular tissue (ITT) transplanted from transgenic donors to wild-type recipient mice. They reported in a previous study that spermatogenesis could be extended from 45 to 85 days when using a polylactide electrospun scaffold. The purpose of the present study was to prepare a structure that is closer to the testicular extracellular matrix and to quantitatively evaluate ITT transplantation and spermatogenesis using bioluminescence imaging, hematoxylin-eosin and immunohistochemical (IHC) staining.

According to the authors, the best spermatogenesis potential was observed at 70 days with fine electrospun scaffolds and spermatogenesis was observed in recipient mice for up to 107 days, about 6 times more than in the “coarse electrospun scaffold” group and the control (without scaffold) group. Their thermal analyses showed that recrystallization occurs during the biodegradation process, which can lead to a more stable microenvironment. The authors conclude that their results can serve as a framework for further research on fertility preservation in prepubertal males.

Although the aim of this study is of interest, the work presented is too preliminary for publication. The results section is not clearly written and the manuscript is not well structured (please see below). Importantly, the conclusions on spermatogenesis are not supported by the results (please see below).

Answer:

Thank you. I do appreciate your valuable comment and improve my study in the long run.

-  Materials and Methods are found in the Results section.

Answer:

Thank you. It is big mistake to let me rephrase the manuscript according your comment.

-  Results are found in the Discussion section.

Answer: Thank you. I rephrased the manuscript according to your comment.

-  The discussion is too long and not a real discussion.

Answer: Thank you. I do my best and has been worked, discussed with my expert colleagues to rephrase the manuscript according to your valuable comment.

-  Many sentences, paragraphs and legends are unclear, confusing or misleading in the manuscript. The English writing should be greatly improved.

Answer: Thank you. I rephrased the manuscript at my best. The English editing has been sent to Online English editing company in Australia to polish my English.

-  The two graphs presented in figure 2 are redundant. Moreover, figures 2 and 3 should be regrouped in the same figure.

Answer: Thank you. I edit the figure 2 and 3 combined as a regrouped in the same figure.

-  The authors used four-week-old transgenic donors and do not justify this choice. Indeed, in prepubertal boys, ITT (in which the only germ cells are spermatogonia) is preserved for fertility preservation. In four-week-old mouse testes however, several types of germ cells can be found: spermatogonia, spermatocytes and spermatids.

Answer: Actually, the beginning of spermatogenesis just server PND. We adopt the reference

Allen, C.M.; Lopes, F.; Mitchell, R.T.; Spears, N. How does chemotherapy treatment damage the prepubertal testis? Reproduction. 2018, 156, R209-R233. The critics also come from the 4-week-old age matched donors and recipients for germ cell proliferation remains at the prepubertal stage at a slow rate in spite of spermatogenesis beginning as early as 1.5 postnatal days [13]. The accommodation for the in vivo BLI tracking system, we were subjected to 4-week-old, which is equal to the pre-pubertal age with germ cell proliferation at a slow rate, functional maturation of Sertoli cells, and the production of functional Leydig cells.

-  Lines 206-208 “Significantly thicker seminiferous tubules in the scaffold groups and a larger seminiferous lumen with significant spermatogenesis than that of ITT without a scaffold”

No measurements of the diameter of seminiferous tubules, of the height of the seminiferous epithelium, of the lumen area and no statistical analyses of these data are shown in this manuscript.

Answer: Thank you. We added a tTble 1 to describe and measure the scale to follow your comment. We do no adopt statistical analysis the diameter, circumstance and area for seminiferous tubules these sections were not always at the same angle.

-  Lines 2018-219 “The HE staining shows significantly adequate spermatogenesis” The different types of germ cells cannot be distinguished at this magnification. Moreover, the percentage of the different types of germ cells (spermatogonia, spermatocytes, spermatids and spermatozoa) per seminiferous tubule should be determined and compared between the different experimental conditions.

Answer: Thank you. The HE staining can only showed roughly the different type of germ cells. The adequate differentiation may relay on full cell maker for spermatogenesis such as DDX4, VASA, MVH, ZBTB 16, PLZF, POU5F1, STAR 8, KIT, SYCP3, DAZL, SALL4, TRA 98, DPPA3 STELLA, 3B-HSD, SOX9, Oct4, and SYCP3. We used part of the cell markers. By the way, we do not calculate statistical analysis for the sperm count from HE staining for the potential insufficiency accurate evidence for the all. But we showed the percentage of different types of germ cells as shown in the Table 1

-  Line 225 “DDX4 represents differentiated germ cells” DDX4 is expressed in all the types of germ cells and not restricted to differentiated germ cells. Other IHC experiments should be performed to detect specifically differentiated germ cells.

Answer: Thank you. The poor quality of IHC staining is not worth of presenting by this quality to mislead reader. We delete this group.

-  Lines 232-233 “The pore size of the fine scaffold showed the best ECM-like niche to engineer ITT transplantation followed by the best spermatogenesis” No quantitative data on the progression of spermatogenesis are shown in the manuscript.

Answer: Thank you. I showed the quantitative data in the Table 1.

-  The same IHC images showing the expression of luciferase are presented in figures 4 and 5.

Answer: Thank you. The expression of luciferase is used as the evidence of graft from donor mice. The expression of luciferase was removed from figure 5 according to your comment.

-  Figure 5: DDX4 staining is more intense in the interstitial tissue, and not within seminiferous tubules as expected.

Answer: Thank you. I agree with your comment. I delete it for not qualified to be published.

-  Figure 5: OCT4-positive cells are not visible at this magnification.

Answer: Thank you. We adjusted magnification x 2 to make it clear.

-  Figure 5: SYCP3 staining is very diffuse and does not seem to specifically stain the spermatocytes.

Answer: Thank you. We adjusted magnification x 2 to make it clear.

-  No quantification and no statistical analyses of the different immunostaining presented in figure 5 are shown.

Answer: Thank you. We did not quantify and measure statistical analysis, instead, just by quality. It should follow our comment.

-  Line 229: “variable degree to achieve spermatogenesis” No quantification and no achievement of spermatogenesis are shown in this manuscript.

Answer: Thank you. We add Table 1 by HE staining. Without full IHC staining for different types of germ cell, it is difficult to quantify concisely.

-  Lines 230-231 “fine scaffold group > course scaffold group > control group” No quantification and no statistical analyses were performed. How can the authors draw this conclusion?

Answer: Thank you. Although we add Table 1, the quantity of photon yield showed in Figure 2 & 3 regrouped that may be another evidence to support the spermatogenesis potential.

-  Lines 244-245 “significantly better spermatogenesis in order as the fine scaffold, coarse scaffold, and control group.” Same comment as above.

Answer: Thank you. Although we add Table 1, the quantity of photon yield showed in Figure 2 & 3 regrouped that may be another evidence to support the spermatogenesis potential.

We rephrased the manuscript to change stronger to better for no full evidence. We confessed that There are several drawbacks to this study. Although BLI signals could represent tissue viability and its observation for spermatogenesis, but not the stages of cell differentiation. The HE staining provides limit information without accuracy to differentiate germ cell types, and our limited cell marker for partial germ cell types of spermatogenesis that is not fully informative to present the real percentage of different types of germ cells. Additional cell markers to analyze the different types of germ cells should be based on germline marker such as DDX4, VASA, MVH, ZBTB 16, PLZF, POU5F1, STAR 8, KIT, SYCP3, DAZL, SALL4, TRA 98, DPPA3 and STELLA. The critics also come from the 4-week-old age matched donors and recipients for germ cell proliferation remains at the prepubertal stage at a slow rate in spite of spermatogenesis beginning as early as 1.5 postnatal days [13]. The accommodation for the in vivo BLI tracking system, we were subjected to 4-week-old, which is equal to the pre-pubertal age with germ cell proliferation at a slow rate, functional maturation of Sertoli cells, and the production of functional Leydig cells [13]. Finally, we did not prove the fertility outcome to produce offsprings from the transplanted ITT. More studies are required toward to produce viable offsprings after the adoption of this research model as a translational study in order to human application.

Author to reviewer:

I am indeed grateful for your detail and professional comments to drive me improvement very much. I will continue these ITT studies by your valuable vision and nice experience. Thank you again.

Chi-Huang Chen MD., PhD

Associate Professor

Taipei Medical University, Taipei, Taiwan

Division Chief of Reproductive Medicine and Director of Reproductive Medicine Center, Taipei Medical University Hospital, Taipei, Taiwan

Reviewer 2 Report

The purpose of this study was to quantify-22 natively to evaluate ITT transplantation and spermatogenesis after application of two scaffolds. The results showed that ITT from 3-week-old mice had the best spermatogenesis potential at 70 days with fine electrospun, and spermatogenesis was observed in recipient mice for up to 107 days, about 6 times more than the coarse electrospun scaffold and control group. The fine electrospun scaffold is closed to the microenvironment of native testicular tissue because it reduces stiffness due to micronization and body fluid infiltration. Thermal analysis showed there is recrystallization during the bodega radiation process, which can lead to the microenvironment becoming more stable. The authors indicated that these results can serve as a framework for further research on fertility preservation in prepubertal males.

Well, the paper seems interesting after reading, but put two many information together, so the readers need to understand and to follow what the authors have been done and what they want to say. Before acceptance for publication, maybe, the manuscript may be needed to write more clearly in order for the readers understand easily. For instance, the Abstract is poorly written, and it should be written better than this.

In summary, this is a interesting manuscript for acceptance in this journal, however, it should be improved for its presentation and format.

Author Response

Response to reviewer 2:

The purpose of this study was to quantify-22 natively to evaluate ITT transplantation and spermatogenesis after application of two scaffolds. The results showed that ITT from 3-week-old mice had the best spermatogenesis potential at 70 days with fine electrospun, and spermatogenesis was observed in recipient mice for up to 107 days, about 6 times more than the coarse electrospun scaffold and control group. The fine electrospun scaffold is closed to the microenvironment of native testicular tissue because it reduces stiffness due to micronization and body fluid infiltration. Thermal analysis showed there is recrystallization during the bodega radiation process, which can lead to the microenvironment becoming more stable. The authors indicated that these results can serve as a framework for further research on fertility preservation in prepubertal males.

Well, the paper seems interesting after reading, but put two many information together, so the readers need to understand and to follow what the authors have been done and what they want to say. Before acceptance for publication, maybe, the manuscript may be needed to write more clearly in order for the readers understand easily. For instance, the Abstract is poorly written, and it should be written better than this.

In summary, this is a interesting manuscript for acceptance in this journal, however, it should be improved for its presentation and format.

Answer

We do appreciate your valuable opinion about our manuscript. We hope that there will be more studies adopting this transgenic mouse model to improve the technology of ITT cryopreservation in the future.

I rephrased the abstract and 30% manuscript to expect the reader to understand easily.

I am grateful for your kindly comment.

Round 2

Reviewer 1 Report

The authors have slightly improved the manuscript. However, even though the authors seem to master bioengineering, the biological part of this manuscript suffers from major drawbacks and is not treated in a rigorous way: the conclusions on spermatogenesis are still not supported by the results. Moreover, parts of the manuscript are still not clear: the English writing still needs to be greatly improved and the manuscript must be proofread very carefully.

Unless the authors deeply change the biological part of the manuscript (by performing additional experiments, adding statistical analyses and showing convincing data) and correct the English writing, this manuscript cannot be accepted for publication.

Major comments:

-   I still don’t understand why 4-week-old transgenic donors were used. The argument presented by the authors is still not clear.

-   Lines 185-208: Luciferase is not expressed under the control of a germ cell-specific promoter. I therefore wonder how the authors can conclude on spermatogenesis by analyzing bioluminescent images. The terms “spermatogenesis … was greater” (lines 193-194), “peak and plateau of spermatogenesis” (lines 195-196), “spermatogenesis gradually decreased” (lines 196-197), “spermatogenesis … was maintained” (line 203), “spermatogenesis was observed” (liens 204-205) cannot be used if these data are not confirmed by histology or immunohistochemistry. Doesn’t bioluminescence reflect the survival of testicular cells rather than spermatogenesis?

-   Lines 225-226: “hematoxylin and eosin (HE) staining showed significantly thicker seminiferous tubules in the scaffold groups and a larger seminiferous lumen with significant better spermatogenesis than that of ITT without a scaffold. (Table 1) ». Since no statistical analyses have been performed, the words “significantly”, “thicker”, “larger”, “significant” and “better” cannot be used.

-   Lines 232-233: “the HE stain showed significantly thicker seminiferous tubules in the scaffold groups and larger seminiferous lumen with significant spermatogenesis than that of ITT without a scaffold” Same comments.

-   Lines 238-239: “The HE stain shows significantly adequate spermatogenesis” The images shown are not magnified enough to see whether or not “spermatogenesis is adequate” (the different germ cell types cannot be distinguished) and the word “significantly” cannot be used since no statistical analyses have been performed.

-   Line 250-251: “variable degree to achieve spermatogenesis” Based on the data obtained by the authors, this conclusion cannot be drawn.

-   Lines 251-252: “the staining results and morphology showed the results of fine scaffold group > course scaffold group > control group (Table 1)” The authors cannot conclude that there are differences between the different experimental conditions without statistical analyses. Moreover, please see my comment on bioluminescence above.

-   Line 255: “followed by the best spermatogenesis (Figure 4)” Based on the data obtained by the authors, it cannot be concluded that the “best spermatogenesis” is obtained with the fine scaffold. Figure 4 only shows the presence of Leydig cells, Sertoli cells, spermatogonial stem cells and maybe meiotic cells (diffuse SYCP3 staining? Nonspecific staining?) after long-term engraftment.

-   Table 1: Secondary spermatocytes are very rarely detected on testicular tissue sections, because meiosis II is a rapid process. However, the authors report the presence of 35% to 50% secondary spermatocytes. I therefore wonder if the authors can identify the different types of germ cells. No enlarged representative images of the different germ cell types are shown in the manuscript for the different experimental conditions.

-   Lines 266-267: “significantly better spermatogenesis in order as the fine scaffold, coarse scaffold, and control group (as shown in Table 1)” The authors cannot conclude that there are differences between the different experimental conditions without statistical analyses.

-   Lines 271-272: “Without statistical analyses for the three groups” The authors therefore cannot conclude that there are differences between the different experimental conditions.

-   Lines 515-516: “The HE staining provides limit information without accuracy to differentiate germ cell types” It is indeed possible to distinguish at high magnification spermatogonia, spermatocytes, spermatids and spermatozoa in HE-stained testicular tissue sections. The results obtained can be further confirmed by immunohistochemistry.

Minor comments:

-   Lines 191-192: There is no such “sperm proliferation”

-   Lines 217-218: “crucial peak between the 42nd and 55th days and then decreases after the 55th day to the end of the 85th tracking day” I don’t see this data on Fig 2.

-   Lines 248: “meiotic cells” instead of “meiosis sperm cells”

-   Line 519: DDX4, VASA and MVH correspond to the same protein

-   Line 519: “STRA 8” instead of “STAR 8”

- In figure legends, columns and lines are confounded.

Author Response

Response to reviewer

Oct 2, 2022

Dear Reviewer                                               

I am grateful for your kindly, informative and strict instruction. We have been objective dependence on BLI as a rapid screen and study strength in vivo. After your valuable and fabulous comments, I learned a lot of knowledge to explore the complex and competitive field related to ITT and spermatogenesis in this small animal model. Coming from your veteran experience, I believe the new study design in a more comprehensive approach shortly. 

General comments:

The authors have slightly improved the manuscript. However, even though the authors seem to master bioengineering, the biological part of this manuscript suffers from major drawbacks and is not treated in a rigorous way: the conclusions on spermatogenesis are still not supported by the results. Moreover, parts of the manuscript are still not clear: the English writing still needs to be greatly improved and the manuscript must be proofread very carefully. 

Unless the authors deeply change the biological part of the manuscript (by performing additional experiments, adding statistical analyses and showing convincing data) and correct the English writing, this manuscript cannot be accepted for publication.

Answer:

With regard to the critical point of view for the evidence on spermatogenesis. Following your valuable comments “It is indeed possible to distinguish at high magnification spermatogonia, spermatocytes, spermatids and spermatozoa in HE-stained testicular tissue sections”, we consulted and invited the co-author, professor Yi-Jen Peng, the Chair of Department of Pathology at National Defense Medical Center, Taipei, Taiwan and professor Yung-Liang Lu whose life long research on mouse spermatogenesis to count the cell number of different germ cell types (200X) to address the manuscript on the evidence-based description instead of too many terms of significant spermatogenesis. The professor Ya‐Li Huang, an expert on statistical analysis, took part in the parameters related to circumference, area of seminiferous tubules and different types of germ cells.

The major discussion highlights the fine PLLA electrospun scaffold we used indicating that a more suitable microenvironment was established. A more appropriate reason for this could be that the micronized PLLA fibers make the overall microenvironment more closely resemble the original testicular tissue. The degradable PLLA also stabilizes the microenvironment due to recrystallization. The quantified BLI   provide the real-time signal on ITT graft condition, not present the spermatogenesis alone between the coarse & fine scaffold and control group. Before ITT engineering through scaffolds can be established as a new approach in ITT grafts and spermatogenesis for the fertility preservation of boys with prepubertal cancer or hematologic disorders, the safety and quality of material sciences must be ensured.

        As for the English writing, we sent this manuscript to IJMS editing with the certificate below:

All the change sentence or words, sentence, paragraph and figure legend will be marked in red.

Major comments:

  1. I still don’t understand why 4-week-old transgenic donors were used. The argument presented by the authors is still not clear.

Answer: Thank you for your comment.

               Our transgenic mice are raised in the SPF (Specific Pathogen Free) area. Since the wild type used in the experiment is purchased from outside, it takes a couple of days of quarantine to enter the SPF. The minimum age of mice that can be purchased is three weeks until experiment on age-matched 4-week-old. Without weaning after 3-week-old, early-weaning manipulation deprives offspring of a certain level of maternal care.

               Given the well-known knowledge, the earlier mouse PND may provide more comprehensive study design but somewhat limitation for in vivo study by age-matched control, it turns out to be subject to study design as such extent.

  1. -   Lines 185-208: Luciferase is not expressed under the control of a germ cell-specific promoter. I therefore wonder how the authors can conclude on spermatogenesis by analyzing bioluminescent images. The terms “spermatogenesis … was greater” (lines 193-194), “peak and plateau of spermatogenesis” (lines 195-196), “spermatogenesis gradually decreased” (lines 196-197), “spermatogenesis … was maintained” (line 203), “spermatogenesis was observed” (liens 204-205) cannot be used if these data are not confirmed by histology or immunohistochemistry. Doesn’t bioluminescence reflect the survival of testicular cells rather than spermatogenesis?

Answer:

Thank you for your comments.

Luciferase is not expressed under the control of a germ cell-specific promoter. The photon yield can only manifest the viable cells, including germ cells and somatic cells. We change the description. This imaging mode, which has become common in preclinical research, is also widely used in many biomedical fields, especially genetic engineering, stem cells, and gene therapy (line 122-124). Our aim was to find a microenvironment more suitable for ITT replantation (line 145-146).

  1. -   Lines 225-226: “hematoxylin and eosin (HE) staining showed significantly thicker seminiferous tubules in the scaffold groups and a larger seminiferous lumen with significant better spermatogenesis than that of ITT without a scaffold. (Table 1) ». Since no statistical analyses have been performed, the words “significantly”, “thicker”, “larger”, “significant” and “better” cannot be used.

Answer: Thank you for your comments.

We paid much attention to measure the circumference and area of seminiferous tubules and calculate the different cell types of germ cells, followed by statistical analysis carefully. There are significant larger circumference and diameter of seminiferous tubules and more spermatid in the find scaffold group than control group. We rephrased the figure legend and paragraphs (Figure 3-5, line 191-220)

  1. -   Lines 232-233: “the HE stain showed significantly thicker seminiferous tubules in the scaffold groups and larger seminiferous lumen with significant spermatogenesis than that of ITT without a scaffold” Same comments.

Answer: Thank you for your comments. Following your instructions, we revised and added Figure 5 (line 194-196, 215-220) accordingly to exclude our previous subjective opinion.

  1.  Lines 238-239: “The HE staining shows significantly adequate spermatogenesis” The images shown are not magnified enough to see whether or not “spermatogenesis is adequate” (the different germ cell types cannot be distinguished) and the word “significantly” cannot be used since no statistical analyses have been performed.

Answer: Thank you for your comments.

        We rephrased this sentence for the different germ cells types for difficult to be distinguished. The description was changed to the existence of spermatogenesis.

  1. Line 250-251: “variable degree to achieve spermatogenesis” Based on the data obtained by the authors, this conclusion cannot be drawn.

Answer: Thank you for your comments.

     We changed the sentence. As a comparison between the fine scaffold, course scaffold, and no scaffold, staining showed all the intact ultrastructure with a variable germ cell types related to spermatogenesis (Figure 6 and 7)(Line 225-226). 

  1. Lines 251-252: “the staining results and morphology showed the results of fine scaffold group > course scaffold group > control group (Table 1)” The authors cannot conclude that there are differences between the different experimental conditions without statistical analyses. Moreover, please see my comment on bioluminescence above.

Answer: Thank you for your comments.

We rephrased this paragraph based on statistical analysis to conclude significant larger circumference and diameter of seminiferous tubules and more spermatid in the find scaffold group than control group. We rephrased the figure legend and paragraphs (Figure 3-5, line 191-220), and as a comparison between the fine scaffold, course scaffold, and no scaffold, staining showed all the intact ultrastructure with a variable germ cell types related to spermatogenesis (Figure 6-7)(Line 224-226). 

  1. -   Line 255: “followed by the best spermatogenesis (Figure 4)” Based on the data obtained by the authors, it cannot be concluded that the “best spermatogenesis” is obtained with the fine scaffold. Figure 4 only shows the presence of Leydig cells, Sertoli cells, spermatogonial stem cells and maybe meiotic cells (diffuse SYCP3 staining? Nonspecific staining?) after long-term engraftment.

Answer: Thank you for your comments.

We rephrased that the drawbacks including additional cell markers which were not able to analyze the different types of germ cells should be based on germline markers such as DDX4, ZBTB 16, PLZF, POU5F1, STRA 8, KIT, SYCP3, DAZL, SALL4, TRA 98, DPPA3 and STELLA (Line 406-413). Sycp3 is expressed in testicular meiotic prophase cells and nuclear staining, usually not in somatic cells. In the figure 7 of our manuscript, the target cells are stained, but the background value is too high, the antibody concentration or action time should be adjusted down, and the background value will decrease. It is a poor processing during Sycp3 staining.

  1. -   Table 1: Secondary spermatocytes are very rarely detected on testicular tissue sections, because meiosis II is a rapid process. However, the authors report the presence of 35% to 50% secondary spermatocytes. I therefore wonder if the authors can identify the different types of germ cells. No enlarged representative images of the different germ cell types are shown in the manuscript for the different experimental conditions.

Answer: Thank you for your comments.

We adopted the figure 3 (60X) as the standard to categorize the different cell types of germ cells, especially spermatogonia, primary spermatocyte, spermatic and spermatozoa to measure and analyze statistically. After check by the pathologist and histologist under 200X magnification to calculate cell number, there are more spermatid in each group.

10 -Lines 266-267: “significantly better spermatogenesis in order as the fine scaffold, coarse scaffold, and control group (as shown in Table 1)” The authors cannot conclude that there are differences between the different experimental conditions without statistical analyses.

Answer: Thank you for your comment.

The comparison between each group was analyzed statistically. We already rephrase as above explanation based on figure 3-4.

  1. -   Lines 271-272: “Without statistical analyses for the three groups” The authors therefore cannot conclude that there are differences between the different experimental conditions.

Answer: Thank you for your comments.

We followed your critical concern on statistical analysis. We consult the expert as co-author to analyze the required comparison between each group.

  1. -   Lines 515-516: “The HE staining provides limit information without accuracy to differentiate germ cell types” It is indeed possible to distinguish at high magnification spermatogonia, spermatocytes, spermatids and spermatozoa in HE-stained testicular tissue sections. The results obtained can be further confirmed by immunohistochemistry.

Answer: Thank you for your comments.

We followed your instruction to distinguish at high magnification (200X) to count the spermatogonia, spermatocytes, spermatids and spermatozoa. Further, we mentioned the drawback including additional cell markers which were not able to analyze the different types of germ cells should be based on germline markers such as DDX4, ZBTB 16, PLZF, POU5F1, STRA 8, KIT, SYCP3, DAZL, SALL4, TRA 98, DPPA3 and STELLA in our study (Line 406-413).

Minor comments:

-   Lines 191-192: There is no such “sperm proliferation”

Answer: We change the sentence “increased BLI intensity” instead of sperm proliferation.

-   Lines 217-218: “crucial peak between the 42nd and 55th days and then decreases after the 55th day to the end of the 85th tracking day” I don’t see this data on Fig 2.

Answer: I am sorry for the misplanted sentence which is found from a previous our published manuscript.

-   Lines 248: “meiotic cells” instead of “meiosis sperm cells”

Answer: We change the description according to your comment.

-   Line 519: DDX4, VASA and MVH correspond to the same protein

Answer: I adopted DDX4 and delete VASA and MVH

-   Line 519: “STRA 8” instead of “STAR 8”

Answer: We corrected the misspelling.

- In figure legends, columns and lines are confounded.

Answer: The figure legends was used row to replace columns.

Round 3

Reviewer 1 Report

The authors greatly improved the manuscript. Minor corrections are necessary:

-  Lines 33-34: “Moreover, spermatogenesis was observed in recipient mice for up to 107 days”. Please replace “spermatogenesis” by “bioluminescent imaging intensity”

-  Lines 193, 202, 206, 208-209, 547-548: “different cell types of germ cells”. Please replace by “different types of germ cells”

-  Lines 195-196: “there is no significant difference in the area and circumference of seminiferous tubules in other germ cell types.” Please clarify this sentence

-  In Figure 4, the number of cells is presented. Is it a number of cells per seminiferous tubule? In the entire grafted testicular tissue?

-  Lines 209-210: “superiority with the percentage of spermatids”. Is it a number of cells or a percentage? If it is a percentage, how was it calculated?

-  Lines 211-212: “there were no significant differences in the percentages of spermatogonia, primary spermatocytes or spermatozoa.” Same comment

-  Lines 410-412: “We do not perform additional cell markers to analyze the different types of germ cells should be based on germline markers such …”. Please rephrase this sentence

Author Response

Response to reviewer

Oct 10, 2022

Dear Reviewer                                               

I am grateful for your minor revision for improving my manuscript.

-  Lines 33-34: “Moreover, spermatogenesis was observed in recipient mice for up to 107 days”. Please replace “spermatogenesis” by “bioluminescent imaging intensity”

Answer: Thank you for your comments.

We followed your comment to replace “spermatogenesis” by “bioluminescent imaging intensity”

-  Lines 193, 202, 206, 208-209, 547-548: “different cell types of germ cells”. Please replace by “different types of germ cells”

Answer: Thank you for your comments.

We followed your comment to change the “different cell types of germ cells”. and replace by “different types of germ cells”

-  Lines 195-196: “there is no significant difference in the area and circumference of seminiferous tubules in other germ cell types.” Please clarify this sentence

Answer: Thank you for your comments.

Previous sentence to address hematoxylin and eosin (HE) staining showed a typical pattern of different types of germ cells of seminiferous tubules (Figure 3), significantly more spermatid, and a larger area and circumference of seminiferous lumen in the fine scaffold group than in the control group (P<0.05) (Figure 4 & 5). The other comparison showed no statistical significance. We rephrased “ In comparison between coarse scaffold group vs find scaffold group and coarse scaffold group vs control group, there is no significant difference in the area and circumference of seminiferous tubules in the other cell types of germ cells.

-  In Figure 4, the number of cells is presented. Is it a number of cells per seminiferous tubule? In the entire grafted testicular tissue?

Answer: Thank you for your comment.

We selected the larger circumference of H&E stain section slide with abundant seminiferous tubules and germ cells at a fix view to count the different types of germ cells under 200X

-  Lines 209-210: “superiority with the percentage of spermatids”. Is it a number of cells or a percentage? If it is a percentage, how was it calculated?

Answer: Thank you for your comments:

“superiority with the percentage of spermatids”. As for the percentage, we misplaced the percentage. Cell number is correct. We initially count the both cell number and percentage. For the statistical analysis of the size of seminiferous tubules and different types of germ cells, the response variable of interest is not normally distributed; we performed the Kruskal–Wallis test to test for differences between treatments and the Nemenyi test for post hoc comparison.

-  Lines 211-212: “there were no significant differences in the percentages of spermatogonia, primary spermatocytes or spermatozoa.” Same comment

Answer: Thank you for your comments.

As for the percentage, we misplaced the percentage. Cell number is correct. We initially count the both cell number and percentage. For the statistical analysis of the size of seminiferous tubules and different types of germ cells, the response variable of interest is not normally distributed; we performed the Kruskal–Wallis test to test for differences between treatments and the Nemenyi test for post hoc comparison.

-  Lines 410-412: “We do not perform additional cell markers to analyze the different types of germ cells should be based on germline markers such …”. Please rephrase this sentence

Answer: Thank you for your comments.

Additional cell markers to analyze the different types of germ cells such as DDX4, ZBTB 16, PLZF, POU5F1, STRA 8, KIT, SYCP3, DAZL, SALL4, TRA 98, DPPA3 and STELLA deserve for further study.